# Polymorphism in IFNλ Can Impact the Immune/Inflammatory Response to COVID-19 Vaccination in Older CMV-Seropositive Adults

**DOI:** 10.3390/vaccines13080785

**Published:** 2025-07-24

**Authors:** Ariane Nardy, Fernanda Rodrigues Monteiro, Brenda Rodrigues Silva, Jônatas Bussador do Amaral, Danielle Bruna Leal Oliveira, Érika Donizetti de Oliveira Cândido, Edison Luiz Durigon, Andressa Simões Aguiar, Guilherme Pereira Scagion, Vanessa Nascimento Chalup, Guilherme Eustáquio Furtado, Marina Tiemi Shio, Carolina Nunes França, Luiz Henrique da Silva Nali, André Luis Lacerda Bachi

**Affiliations:** 1Post-Graduation Program in Health Sciences, Santo Amaro University (UNISA), São Paulo 04743-030, SP, Brazil; arianenardy@estudante.unisa.br (A.N.); monteiro.fernanda13@gmail.com (F.R.M.); brendarodrigues@estudante.unisa.br (B.R.S.); mtshio@prof.unisa.br (M.T.S.); cnfranca@prof.unisa.br (C.N.F.); lnali@prof.unisa.br (L.H.d.S.N.); 2ENT Research Laboratory, Department of Otorhinolaryngology—Head and Neck Surgery, Federal University of São Paulo (UNIFESP), São Paulo 04021-001, SP, Brazil; jbamaral@unifesp.br; 3Hospital Israelita Albert Einstein, São Paulo 05652-900, SP, Brazil; danielle.durigon@einstein.br; 4Department of Microbiology, Institute of Biomedical Sciences, University of São Paulo, São Paulo 05508-000, SP, Brazil; erikacandido@usp.br (É.D.d.O.C.); eldurigo@usp.br (E.L.D.); andressa.simoes@icb.usp.br (A.S.A.); gui.scagion@gmail.com (G.P.S.); vmnchalup@usp.br (V.N.C.); 5Polytechnic University of Coimbra (IPC), Rua da Misericórdia, Lagar dos Cortiços—S. Martinho do Bispo, 3045-093 Coimbra, Portugal; guilherme.furtado@ipc.pt; 6SPRINT—Sport Physical activity and health Research & Innovation Center, Polytechnic University of Coimbra, Rua Dom João III—Solum, 3030-329 Coimbra, Portugal; 7Center for Studies on Natural Resources, Environment, and Society (CERNAS), Polytechnic University of Coimbra, Bencanta, 3045-601 Coimbra, Portugal

**Keywords:** SARS-CoV-2, immunosenescence, inflammaging, immunoglobulins, cytokines, immunophenotyping

## Abstract

Background: Chronic cytomegalovirus (CMV) infection may favor the development of immunosenescence and inflammation that impair vaccine responses, including COVID-19. In addition, the polymorphism of the interferon-lambda gene (IFNλ) affects COVID-19 immune responses in older adults. Objective: We aimed to investigate the impact of IFNλ polymorphism (*IL28B* gene-rs12979860) on the immune/inflammatory response to vaccination with CoronaVac for COVID-19 in older adults who were CMV-seropositive. Methods: Blood samples from 42 CMV-seropositive older adults (73.7 ± 4.5 years) were collected before and 30 days after immunization with a second dose of the CoronaVac vaccine to evaluate the immune/inflammatory response. Results: At genotyping, 20 subjects were homozygous for the C/C alleles (Allele-1 group), 5 were homozygous for the T/T Alleles (Allele-2 group), and 17 were heterozygous (C/T, Alleles-1/2 group). The Allele-1 group showed higher IgG levels for COVID-19 (*p* = 0.0269) and intermediate monocyte percentage (*p* = 0.017), in contrast to a lower non-classical monocyte percentage (*p* = 0.0141) post-vaccination than pre-vaccination. Also, this group showed that IgG levels for CMV were positively associated with a systemic pro-inflammatory state and senescent T cells (CD4+ and CD8+). The Allele-2 group presented higher IFN-β levels at pre- (*p* = 0.0248) and post-vaccination (*p* = 0.0206) than the values in the Allele-1 and Alleles-1/2 groups, respectively. In addition, the Allele-2 and Alleles-1/2 groups showed that IgG levels for COVID-19 were positively associated with a balanced systemic inflammatory state. Conclusion: CMV-seropositivity in older adults who had Allele-1 could lead to an unbalanced systemic inflammatory state, which may impair their antibody response to COVID-19 vaccination compared to other volunteer groups.

## 1. Introduction

The coronavirus 2019, or COVID-19, which emerged from the novel coronavirus that causes severe acute respiratory syndrome (severe acute respiratory syndrome coronavirus-2-SARS-CoV-2), was first identified in Wuhan, China, in December 2019 and spread rapidly around the world, evolving into a pandemic [1]. Although the global emergency is under control, some important observations have been noted, including that older adults are one of the populations most affected by COVID-19, especially due to immunosenescence, a multifactorial and dynamic phenomenon associated with the decline in immune responses with age [2,3,4].

Immunosenescence is known to involve significant changes in the activity, number, and profile of immune cells, especially monocytes [5], a heterogeneous type of circulating leukocyte with phenotypic and specific characteristics. In this sense, these cells are categorized into three subtypes, based on the expression of the protein surfaces CD14 and CD16. In general, it is reported that classical monocytes (CD14++/CD16−) not only represent the major circulating monocyte subtype (80–90%) but also show a remarkable ability to migrate into tissues, whilst intermediate monocytes (CD14++/CD16+) represent the lowest circulating monocyte type (~5%), but are considered the most inflammatory due to their ability to secrete large amounts of pro-inflammatory cytokines, and non-classical monocytes (CD14+/CD16++) make up 5–10% of circulating monocytes and are generally considered to have both anti-inflammatory properties and to be patrolling cells due to their ability to clear debris [5,6].

In addition to these aspects, another notable feature of immunosenescence is the reduction in the number and proliferative capacity of naïve T lymphocytes, which decreases the ability to generate immune responses to novel antigenic challenges, such as vaccination, in contrast to the accumulation of effector and memory T lymphocytes [3,7]. It has also been reported that alterations in the phenotype of circulating T lymphocytes towards more differentiated stages or even a senescent profile due to the expression of CD57 in conjunction with a reduced proliferative capacity due to the loss of expression of co-stimulatory molecules, particularly CD28, are often associated with a lower immune response to vaccination in older adults [8,9].

It is important to emphasize that aging can also induce systemic, sterile, chronic low-grade inflammation, a phenomenon termed “inflammaging”, which has been identified as a pillar of increased risk of morbidity and mortality in this population [4]. Features of this phenomenon include increased release of pro-inflammatory cytokines, particularly from immune cells such as monocytes and lymphocytes, and a progressive decline in the ability of the immune system to respond efficiently to infection and vaccination [10].

According to the literature, both immunosenescence and inflammaging can be affected by infection with cytomegalovirus (CMV), a human β-herpesvirus that is ubiquitous in the population [11] and generally remains latent in immune cells. It is of paramount importance to mention that monocytes represent one of the main circulating cell types that could be infected and a reservoir for CMV, as well as favor viral reactivation through their differentiation [12]. Over time, repeated reactivations and persistent infection of this virus can lead to an immunologic risk profile that includes an increase in the frequency of terminally differentiated T cells (CD57+), an inverted ratio of CD4+/CD8+ T cells, an abnormal expansion of CMV-reactive CD8+ T cells, and a reduction in the number of B cells [13]. Moreover, pro-inflammatory cytokines are involved in CMV reactivation, so repeated cycles of asymptomatic CMV reactivation may contribute to the increase in systemic pro-inflammatory state in the older adult population [13,14].

Regarding the host response to viral infection, a group of cytokines with distinct antiviral activities, such as interferon (IFN), is able to trigger the transcription of hundreds of genes in host cells that are involved in both direct combat and in shaping the subsequent immune response against viral infection [15]. Thus, reduced production of type-I IFN, such as IFN-α and IFN-β, observed in monocytes of older adults may increase susceptibility to respiratory viral infections, including SARS-CoV-2, as these IFNs are central to both host resistance and viral defense [16]. Moreover, type-III IFN, also known as IFN-lambda (IFNλ), can generate host resistance to viral infections by activating multiple intracellular antiviral effectors and stimulating the adaptive immune response [16,17]. In particular, in CMV infections, polymorphisms in the genomic region of IFNλ 3/4 have been reported to be associated with greater susceptibility to the replication of this virus [18]. Additionally, the same polymorphism has been associated with the age and severity of COVID-19 infection [19].

Taken together, these data suggest that the polymorphism in IFNλ 3/4 may play an important role in both CMV and SARS-CoV-2 infections. Corroborating this suggestion, our group has recently shown that the polymorphism in the gene IFNλ 3/4 (rs12979860 C/T), more specifically, the presence of homozygosity for allele 1 (C/C), negatively affects the antibody response not only to SARS-CoV-2 infection but also to immunization with Oxford-AstraZeneca vaccine, known as ChAdOx-1, an adenovirus vector vaccine, in CMV-seropositive older adults [20].

Although these data are interesting, the impact of IFNλ 3/4 polymorphism on the immune/inflammatory response of CMV-seropositive older adults immunized with different types of COVID-19 vaccines is still unclear. Therefore, in the present study, we aimed to expand our knowledge on the impacts of the IFNλ 3/4 gene polymorphism (rs12979860 C/T) on the immune/inflammatory responses of CMV-seropositive older adults immunized against COVID-19 with CoronaVac, an inactivated vaccine containing the entire SARS-CoV-2 virus.

## 2. Materials and Methods

### 2.1. Design of the Study

In this prospective observational study, with two time points (before and after) of outcome analysis, which followed the STROBE (Strengthening the Reporting of Observational Studies in Epidemiology) guideline to ensure methodological rigor, 42 older adults of both sexes (30 women and 12 men) aged between 60 and 85 years were enrolled.

### 2.2. Recruitment of Participants

All participants were recruited and selected from the Discipline of Geriatrics and Gerontology of the Federal University of São Paulo (UNIFESP), with the assistance of a geriatric physician who collaborated in the study.

### 2.3. Selection Criteria

The inclusion criteria were as follows: (i) age between 60 and 85 years; (ii) seropositive for CMV; (iii) seronegative for HIV; (iii) no neurological, liver, or kidney disease and cancer; (iv) no flu-like symptoms during the study period; (v) completion of the schedule of two doses of the CoronaVac vaccine; and (vi) consent to participate in the study. The exclusion criteria were as follows: (i) corticosteroid therapy, (ii) use of other anti-inflammatory drugs, (iii) convalescent plasma therapy for at least 2 prior to the start of the study and in any phase of the present study, and (iv) refusal to provide a blood sample. It is important to clarify that the same geriatric physician carefully examined and clinically supervised all subjects during the development of the study.

Sample size and statistical power were estimated based on the ANOVA test with an effect size of 0.30, at a level of 0.05, level and statistical power of 0.95. Moreover, a minimum of 41 individuals was assumed based on our previous study [20]. In this context, of the 67 older adults invited to voluntarily participate in the study, 25 were excluded because they had received the COVID-19 vaccination with other vaccines. Therefore, the number of older adults (*n* = 42) vaccinated with the CoronaVac vaccine reached the estimated sample size.

### 2.4. Ethical Aspects

Before inclusion, all participants were informed in detail about the study procedures, and those who volunteered to participate gave their written consent by signing the Free and Informed Consent Form, previously approved by the Research Ethics Committee of the Universidade Santo Amaro (UNISA) under protocol number 4.350.476. The study respected and complied with the Declaration of Helsinki [21] and guidelines of Resolution No. 466/2012 of the National Health Council [22].

### 2.5. Biological Sample Collection

Fasting peripheral blood samples were collected in tubes containing or not EDTA anticoagulant at two different time points: before and 30 days after administration of the second dose of CoronaVac vaccine for COVID-19.

Aliquots of serum (at least 500 µL) were obtained after blood clotting in the collection tubes and subsequent centrifugation of the samples at 800× *g* for 10 min at 4 °C. The aliquots were then frozen at −80 °C until the concentrations of specific IgG for the SARS-CoV-2 and CMV antigens and of pro- and anti-inflammatory cytokines were determined, as described below.

The blood sample collected in tubes containing anticoagulant EDTA was used to obtain peripheral blood mononuclear cells (PBMCs). In brief, the blood was mixed 1:1 with PBS (1×, pH = 7.4) and transferred to a Falcon^®^ tube containing Ficoll-Hypaque (GE Healthcare Bio-Sciences AB, Uppsala, Sweden). After centrifugation, 1 × 10^6^ PBMCs were mixed with 1 mL of freezing medium (10% DMSO plus 90% fetal bovine serum) and stored in liquid nitrogen until genotyping and immunophenotyping, as described below.

#### 2.5.1. Determination of Specific IgM and IgG for CMV Antigens

Serum concentration of specific IgM and IgG for CMV antigen was determined using a commercial ELISA kit (BioClin, Minas Gerais, Brazil) according to the manufacturer’s instructions. Standards and positive/negative controls were prepared according to the manufacturer’s instructions, and subjects who presented values ≥ 1.1 for IgM specific to p52 antigen, or ≥1.32 for IgG specific to glycoprotein B, were classified as CMV-seropositive.

#### 2.5.2. Determination of Specific IgG for SARS-CoV-2 Antigens

The serum concentration of specific IgG for SARS-CoV-2 antigens was determined by an “in-house” ELISA assay according to the method previously described by our group [23]. In brief, microplates (96 wells) were coated with nCoV-PS-Ag7 (Fapon Biotech Inc., Dongguan, China) at a concentration of 0.12 µg/mL containing the nucleoprotein (N), membrane (M), and spike (S) antigens. After blocking non-specific sites, serum samples were added, diluted in 0.1% PBS-Tween buffer with 1 M NaCl, and incubated at 37 °C for 2 h. After washing the wells, the peroxidase-conjugated anti-human IgG (1:10,000) was added and incubated at 37 °C for 1 h. After washing again, the reaction was developed with TMB solution (3,3′,5,5′-tetramethylbenzidine) and then stopped with H_2_SO_4_ solution (2N). The absorbance was measured at 450 nm in a microplate reader (Multiskan Sky Microplate Spectrophotometer, Thermo Fisher Scientific, Waltham, MA, USA).

#### 2.5.3. Analysis of the Neutralizing Antibodies

SARS-CoV-2 virus (GenBank: MT350282) was used to perform the virus neutralization test (VNT) based on the cytopathic effect (CPE), as previously described [24]. In brief, serum samples were diluted (from 1:20 to 1:2560), mixed with virus in equal volumes (100 infectious tissue culture doses, 100% endpoint per well—VNT100), and pre-incubated at 37 °C for 1 h for virus neutralization. After incubation, the mixtures containing serum and virus were transferred to a cell culture of Vero cells (ATCC CCL-81) previously prepared in 96-well microplates (5 × 10^4^ cells/mL) and incubated for 72 h at 5% CO_2_ and 37 °C in a biosafety level 3 (BSL3) laboratory. The presence or absence of CPE-VNT against both the Wuhan reference strain and the delta variant, which was prevalent in Brazil at the time of sampling, was determined by optical microscopy. The neutralizing antibody titer was determined as the highest serum dilution capable of neutralizing the viral cytopathic effect. A reference serum from an RT-qPCR-positive individual who had a CPE-VNT > 640 was used as a positive control in each test. It is also worth noting that subjects who had at least a 2-fold increase in titers of the VNT were named ”responders” to COVID-19 vaccination.

#### 2.5.4. Determination of Circulating Cytokine Concentrations

Serum concentrations of the cytokines IL-6, IL-10, IL-17, IFN-α2, IFN-β, IFN-γ, and TNF-α were determined using the LEGENDplex™ bead-based multiplex assay (Biolegend, San Diego, CA, USA). In brief, serum samples were first diluted 2-fold, and 25 µL of the diluted sample was used for the assay. According to the manufacturer’s instructions, the concentration of each cytokine was calculated using the respective standard curve, which had a correlation coefficient of 0.94 to 0.99, an intra-assay coefficient of variance of 3–4%, and an inter-assay coefficient of variance of 7–9%. In addition, the linearity of all parameters evaluated was in the range of 0–10,000 pg/mL. This analysis was performed using the BD Accuri™ C6 Plus flow cytometer (BD Biosciences, San Jose, CA, USA), and data were analyzed using LEGENDPlex™ V8.0 software (Biolegend, San Diego, CA, USA).

#### 2.5.5. Determination of Viral Load of CMV

In order to determine the viral load of CMV in serum samples, the viral DNA was first extracted using the PureLink Genomic DNA Mini Kit (Thermo Fisher Scientific, Waltham, MA, USA), which consists of a kit based on a silica column. After extraction, all samples were subjected to real-time PCR (RT-PCR) with primers complementary to the endogenous GAPDH gene: forward primer 5′-CACCCACTCCTCCACCTTTGAC and reverse primer 5’-AGACCATCCACCTCCACTTCTC [25]. For this amplification, the Sybr Green platform was used, and the cycling conditions were 95 °C for 10 min, followed by 40 cycles of 95 °C for 15 s, and 60 °C for 1 min. A melting curve was then generated to distinguish the amplified products from the positive control. Samples were considered positive if the RT-PCR showed a major peak at 77 °C ± 1 °C. All samples were positive for this gene. Once the presence of DNA in the extracted samples was ensured, they were analyzed for the presence of CMV by real-time PCR using the TaqMan platform. The primers used were forward 5′-CCTGGTGGACTATGCTTAATG and reverse 5’-GGAAGCAGCAATGTCGTAGTACAAT, and the probe used was FAM-ATTCTCATGGGAGCTTTT-BHQ [26]. The cycling conditions were 95 °C for 10 min, followed by 40 cycles of 95 °C for 15 s and 60 °C for 1 min. In all reactions, the final primer concentration was 3.2 pmol, and the probe concentration was 2.5 pmol per reaction. All PCR assays were performed on a Step One Plus Real-Time PCR equipment (Applied Biosystems, Thermo Fisher Scientific, Waltham, MA, USA).

#### 2.5.6. Analysis of the Polymorphism in the IL28B Gene

Peripheral blood mononuclear cells (PBMCs), previously stored, were thawed, and DNA was extracted using the PureLink® Genomic DNA Mini Kit (Thermo Fisher, Waltham, MA, USA), according to the manufacturer’s instructions. DNA from each sample was quantified in a NanoDrop Microvolume Spectrophotometer One (Thermo Fisher, Waltham, MA, USA) to determine the efficiency and concentration of DNA extraction. Then, genotyping for polymorphism in the *IL28B* gene (rs12979860 C/T) was performed using the TaqMan platform to discriminate alleles (primers and probes designed by Invitrogen, Thermo Fisher, Waltham, MA, USA), and analyses were carried out in StepOne Plus software version 2.3, as described by Prokunina-Olsson and colleagues [27]. Subsequently, approximately 4 ng of DNA from each sample was used to perform genotyping assays by RT-PCR amplification steps with specific probes for each allele, following previously published protocols [20,27]. The reaction was performed on the StepOnePlus™ Real-Time PCR System platform (Thermo Fisher, Waltham, MA, USA). The results obtained in the assessment of allelic discrimination of genotyping for the polymorphism in IFNλ (*IL28B* gene) are shown in Appendix A.

#### 2.5.7. Immunophenotypic Characterization of Monocyte Subtypes and T Cells

Previously stored PBMCs (1 × 10^6^ cells per vial) were thawed and then incubated with monoclonal antibodies conjugated to the fluorochromes anti-CD14-FITC and anti-CD16-PE or anti-CD4 FITC, anti-CD8 APC, anti-CD57 PE, and anti-CD28 PercP, and isotype controls (all from BD Bioscience) for 30 min at room temperature. After this incubation period, the cells were washed and resuspended in PBS. The percentage of monocyte subtypes (classical, intermediate, and non-classical) and naïve, double-positive, and senescent T cells was determined in a FACSCalibur flow cytometer (Becton Dickinson Immunodiagnostic Systems, San Jose, CA, USA) at a minimum number of 10,000 events. Data analysis was performed using BD CellQuest Pro software, version 5.1 (Becton Dickinson Immunodiagnostic Systems, San Jose, CA, USA). Appendix A illustrates the strategy used in the study to define monocyte subtypes (A, non-classical—CD14+CD16++; intermediate—CD14++CD16+, and classical—CD14++CD16−) and CD4+ T cells (B, naïve—CD4+CD28+CD57−; double-positive—CD4+CD28+CD57+, and senescent—CD4+CD28−CD57+).

### 2.6. Statistical Analysis

First, the normality of the data and the homogeneity of the variances were checked using the Shapiro–Wilk test and the Levene test.

Parametric data (demographic and anthropometric data, as well as ratio between pro- and anti-inflammatory cytokines) were expressed in means and standard deviations (X ± SD), and comparison between subject groups was assessed by the Student *t*-test (intragroup analysis) or one-way ANOVA with Tukey post hoc test (intergroup analysis). As other data were considered non-parametric variables, results were expressed as median and interquartile ranges (X_IR). Both the Wilcoxon test (Intragroup analysis) and the Kruskal–Wallis test with the Mueller–Dunn post-test (intergroup analysis) were used to determine significant differences between volunteer groups and time points. In addition, the Spearman correlation coefficient test was also used.

For all analyses, *p* ≤ 0.05 was considered statistically significant, and the GraphPad Prism 8.1.2 software was used.

## 3. Results

### 3.1. Characterization of the Volunteer Group

Table 1 shows the data on age, number of participant subjects (total and separated by sex), and anthropometry of the individuals who participated in the present study, both overall and grouped by allelic difference. It is important to clarify that all subjects who participated in this study completed the vaccination schedule, which included two doses of the CoronaVac vaccine. No differences were found in these parameters between the volunteer groups.

At this point, it is also of utmost importance to mention that, in accordance with the inclusion criteria, all subjects presented levels of IgG-specific for CMV, not IgM, above the cut-off described in the manufacturer’s instructions; thus, they were classified as CMV-seropositive. In addition, it is worth noting that none of the volunteers exhibited CMV reactivation at the time points studied here, either before or after COVID-19 vaccination (data not shown).

### 3.2. Specific IgG for the CMV and SARS-CoV-2 Antigens in the Volunteer Cohort

Figure 1A,B show the results of the analysis of serum levels of specific IgG for SARS-CoV-2 and CMV antigens, both before (pre) and 30 days after administration of the second dose of CoronaVac vaccines (post), in the volunteer group. Higher serum levels of specific IgG for SARS-CoV-2 (Figure 1A, *p* = 0.0392) were found after (post) than before vaccination (pre), in contrast to the lower serum levels of specific IgG for CMV (Figure 1B, *p* = 0.0394). It is also important to point out that only 33% of volunteers showed an increase in serum IgG levels for SARS-CoV-2 after vaccination.

### 3.3. Specific IgG for the SARS-CoV-2 and CMV Antigens in the Volunteer Cohort Grouped According to Allelic Discrimination

Figure 2A,B show the data obtained by analyzing the serum levels of specific IgG for SARS-CoV-2 and CMV antigens, both before (pre) and 30 days after the administration of the second dose of CoronaVac vaccines (post), when the volunteers were categorized according to allelic discrimination into Allele-1 (with only cytosine in the allele pair of alleles, C/C), Allele-2 (with only thymine in the allele pair of alleles, T/T), and Alleles-1/2 (with both cytosine and thymine in the allele pair of alleles, C/T). Higher serum IgG levels for SARS-CoV-2 antigens were found post-vaccination than pre-vaccination, specifically in the Allele-1 group (Figure 2A, *p* = 0.0269). Similarly to what was previously mentioned, it is important to highlight that 35% of subjects in the Allele-1 group, 40% in the Allele-2 group, and 35% in the Alleles-1/2 group showed an increase in the serum specific IgG levels for SARS-CoV-2 antigens post-vaccination.

Beyond these data, it is important to point out that the neutralizing antibodies were assessed through the VNT, and it was observed an increase, of at least 2-fold, in their titers in 7 of 20 volunteers (35%) in the Allele-1 group, in 2 of 5 volunteers (40%) in the Allele-2 group, and in 5 of 17 volunteers (~29.5%) in the Alleles-1/2 group (Appendix A). In addition, Figure 2C,D show the results obtained when evaluating the total serum levels of specific IgG for SARS-CoV-2 and CMV antigen in the volunteer group were not only categorized according to allelic discrimination but also grouped into non-responders (NRE, those without increase in the neutralizing antibody titers) and responders (RE, those with an increase in the neutralizing antibody titers) according with the results obtained in the VNT. No statistically significant differences were found in the total serum levels of specific IgG for SARS-CoV-2 and CMV antigens between pre- and post-vaccination time points. In view of the fact that the volunteer groups showed a similar number of subjects with an increase in the neutralizing antibody titers post-vaccination, as well as that we were unable to evidence any significant differences in the evaluation of serum levels of specific IgG for SARS-CoV-2 and CMV antigens, in conjunction with the small number of volunteers in the Allele-2 subgroup (responders’ subgroup), which would make statistical analysis impossible, we have opted to present the following results with the volunteer groups without this separation into responders and non-responders.

### 3.4. Immunophenotyping of T Cells and Monocyte Subtypes in the Subject Cohort Grouped According to Allelic Differentiation

Figure 3 shows the percentages of naïve (CD28+CD57−, Figure 3A,D), double-positive (CD28+CD57+, Figure 3B,E), and senescent (CD28−CD57+, Figure 3C,F) CD4+ T cells and CD8+ T cells, respectively, as well as the percentages of monocyte subtypes [non-classical (CD14+CD16++, Figure 3G), intermediate (CD14++CD16+, Figure 3H), and classical (CD14++CD16−, Figure 3I)], both before and 30 days after administration of the second dose of CoronaVac vaccine in volunteers categorized into Allele-1 (C/C), Allele-2 (T/T), and Alleles-1/2 (C/T) groups according to allele differentiation. Post-vaccination, a lower proportion of non-classical monocytes (Figure 3G) was observed in both the homozygous group for Allele-1 (*p* = 0.0141) and the heterozygous group (Alleles-1/2, *p* = 0.007) than the values pre-vaccination. Interestingly, an increase in the proportion of intermediate monocytes after vaccination was only observed in the subjects with homozygosity for allele 1 compared to pre-vaccination values (*p* = 0.017, Figure 3H). No other significant difference was found.

### 3.5. Systemic Inflammatory Status in the Volunteer Cohort Grouped Based on the Allelic Discrimination

Figure 4 shows the results obtained in the evaluation of serum concentrations of pro- and anti-inflammatory cytokines (Figure 4A–G), both before and 30 days after the administration of the second dose of CoronaVac vaccine in the volunteers categorized according to allelic differentiation into groups Allele-1 (homozygous, C/C), Allele-2 (homozygous, T/T), and Alleles-1/2 [heterozygous, C/T)]. Higher IFN-β serum levels (Figure 4F) were observed in the individuals in Allele-2 (T/T) group not only at the pre- but also at the post-vaccination time points, than the levels observed both in volunteers of Allele-1 group (C/C, *p* = 0.0248 and *p* = 0.0206, respectively) and in the Alleles-1/2 group (C/T, *p* = 0.0305 and *p* = 0.0338, respectively).

### 3.6. Analysis of the Ratio Between Pro- and Anti-Inflammatory Cytokines in the Volunteer Cohort According to Allelic Discrimination

Figure 5 shows the ratio between pro-inflammatory cytokines and the anti-inflammatory cytokine IL-10 (Figure 5A–F), both before and 30 days after administration of the second dose of the CoronaVac vaccine in the volunteers categorized according to their allelic differentiation into groups Allele-1 (homozygous, C/C), Allele-2 (homozygous, T/T), and Alleles-1/2 (heterozygous, C/T). A significant reduction in the IL-6/IL-10 ratio (Figure 5A, *p* = 0.0214) was observed in subjects who were homozygous for Allele-2 (T/T) post-vaccination time point compared to the values observed pre-vaccination. In addition, higher IFN-β/IL-10 ratio values (Figure 5E) were observed in the Allele-2 group (homozygous, T/T) both pre- and post-vaccination than in the Allele-1 group (homozygous, C/C, *p* = 0.0050 and *p* = 0.0009, respectively) and the Alleles-1/2 group (heterozygous, C/T, *p* = 0.0020 and *p*¡0.001, respectively).

### 3.7. Correlation Analysis Between Immune/Inflammatory Parameters Assessed in the Volunteer Cohort Grouped Based on the Allelic Discrimination

Table 2 shows the significant results of the Spearman correlation coefficient analysis with the specific IgG levels, both for SARS-CoV-2 and CMV antigens, as the main parameters studied. Only the group with Allele-1 showed that serum IgG levels for SARS-CoV-2 antigens (IgG-COVID-19) had a negative correlation with BMI values, especially at the time before vaccination, whereas serum IgG levels for CMV antigens (IgG-CMV) were positively associated not only with systemic pro-inflammatory status but also with the presence of senescent T cells (both CD4+ T and CD8+ T cells) at the post-vaccination time point, regardless of the time point studied. Concerning the Allele-2 group, at the pre-vaccination time point, negative correlations were found between serum IgG levels for SARS-CoV-2 antigens (IgG-COVID-19) with Il-17 and TNF-α or IL-17/IL-10 and TNF-α/IL-10 ratios, in contrast to positive correlations between the same antibodies and the circulating levels of IFN-α2 and IFN-γ, as well as the IFN-α2/IL-10 and IFN-γ/IL-10 ratios, whereas, at the post-vaccination time point, the serum IgG levels for SARS-CoV-2 antigens (IgG-COVID-19) showed both a positive correlation with circulating IL-10 levels and a negative correlation with the percentage of senescent CD8+ T cells. Lastly, the Alleles-1/2 group (heterozygous) showed only a negative correlation between serum IgG levels for SARS-CoV-2 antigens (IgG-COVID-19) and the percentage of intermediate monocytes at the pre-vaccination time point and positive correlations between the systemic levels of these antibodies (IgG-COVID-19) and the circulating levels of IL-10 and IFN-γ, at the post-vaccination time point.

### 3.8. Multivariate Linear Regression Analysis Between Immune/Inflammatory Parameters Assessed in the Volunteer Cohort Grouped Based on the Allelic Discrimination

In addition, we performed a multivariate linear regression analysis adjusted for serum IgG-specific levels for SARS-CoV-2 antigens (Table 3). This analysis showed that the Allele-1 group showed a significant effect of antibody response to COVID-19 vaccination both before and after vaccination in BMI values and, at the post-vaccination time point, in systemic levels of IL-17 and IFN-γ, and also in the ratios of IL-17/IL-10 and IFN-γ/IL-10. In the Allele-2 group, at the pre-vaccination time point, a significant effect was observed in IgG-CMV levels, whereas, at the post-vaccination time point, significant effects in IgG-CMV levels and the non-classical monocyte percentage were observed. Lastly, the Alleles-1/2 (heterozygous) group showed a significant effect on age only at the post-vaccination time point.

## 4. Discussion

In the present study, it was shown, for the first time, that older adults who are CMV-seropositive, grouped according to the polymorphism of the IFNλ 3/4 gene (rs12979860 C/T), showed a different immune/inflammatory response after immunization with two doses of the CoronaVac vaccine for COVID-19. First of all, it is paramount to mention that none of the volunteers presented CMV reactivation in the time points studied here, and also they showed a significant increase in specific IgG levels in serum for SARS-CoV-2 in conjunction with a decrease in IgG levels for CMV post-vaccination. Although these results may suggest that the subjects had reached a favorable immunological state that allowed them to respond to COVID-19 vaccination, there is no consensus that CMV seropositivity could impair or not the immune response to vaccination, especially in older adults [28,29]. In this context, the literature suggests, among other things, that mild chronic inflammation could be a corollary factor, as it can act as a “double-edged sword” by enhancing or weakening the immune system of older adults to antigenic challenges, such as vaccination [29].

Despite these results, showing that two doses of the CoronaVac vaccine were able to induce a satisfactory immune response, it is also important to mention that the low immunogenicity observed here [both when the volunteers were assessed in conjunction (33.3%), as well as when they were categorized into Allele-1 (35%), Allele-2 (40%), and Alleles-1/2 (35%) groups] is consistent with the literature [30]. Although one study reported that immunogenicity in older subjects was 70.37% after a 4-week administration of the second dose of CoronaVac, it was also pointed out that pre-existing antibodies against SARS-CoV-2 antigens may interfere with the subsequent response to repeated antigenic stimulation [31], as may be the case in our cohort of volunteers.

As mentioned above, IFNs, particularly IFNλ, have a complementary effect on viruses, including CMV and SARS-CoV-2 infection. In this context, although it has been demonstrated that the lower viral clearance observed in children with acute upper respiratory tract infections and in adult/older SARS-CoV-2-infected individuals may be associated with homozygous–homozygous variants of the IFNλ3/4 gene [32], the impact of the IFNΛ3/4 gene polymorphism on COVID-19 vaccination, especially in the older adult population, is still poorly understood. Therefore, we analyzed the SNPs of the IL-28 gene SNPs (rs12979860 C/T), a type III-like IFN [33], and it was evidenced that 20 individuals (47.6%) presented only cytosine in the allele pair of alleles (C/C, Allele-1 group), whereas 5 (11.9%) had only thymine (T/T, Allele-2 group), and 17 (40.5%) had both cytosine and thymine (C/T, Alleles-1/2 group), in the 3Kpb upstream region of the *IL28B* gene [34]. These data confirm our previous study in which the number of older adults who had genotypes with cytosine (C/C and C/T) was higher than thymine (T/T) only [20].

Apart from these observations, the evaluation of the IgG response to COVID-19 vaccination in the subject groups showed a significant increase in serum-specific IgG levels for SARS-CoV-2 antigens only in the Allele-1 group, as well as a higher percentage of intermediate monocytes and a lower percentage of non-classical monocytes after vaccination than before vaccination. In addition, before COVID-19 vaccination, the serum IgG-specific levels for SARS-CoV-2 antigens showed a significant negative correlation with BMI values, and a systemic pro-inflammatory status was positively associated with serum levels of specific IgG for CMV antigens, both before and after COVID-19 vaccination. Interestingly, the multivariate regression analysis considering IgG-specific serum levels for SARS-CoV-2 antigens not only confirmed the negative influence of increased BMI but also showed a significant impact of circulating levels of IL-17 and IFN-γ, and their respective ratios (IL-17/IL-10 and IFN-γ/IL-10). Regarding the antibody response to COVID-19 infection and vaccination in older adults who are CMV-seropositive and were categorized according to the IFNλ 3/4 gene polymorphism, we have previously shown that the older individuals who had allele 1 had higher IgG-specific levels for SARS-CoV-2 antigens than the others volunteer groups only 8 months after SARS-CoV-2 infection, and we hypothesized that this result was related to the negative correlation between IgG-specific levels for SARS-CoV-2 and CMV seropositivity observed in the Allele-1 group [20].

In light of these previous findings, we can putatively suggest that the elevated IgG-specific levels for SARS-CoV-2 antigens found exclusively in the Allele-1 group may be related to a differential induction of the immune response, which, although not showing a direct correlation with CMV seropositivity, even so, could be influenced by CMV infection, as these subjects showed a significant positive correlation between a systemic pro-inflammatory status and CMV seropositivity. In this context, it is generally recognized that chronic CMV infection, which is closely related to the different cycles of reactivation of this virus throughout life, leads to an increase in the production of pro-inflammatory cytokines and thus affects systemic inflammation and may also impair the vaccination response [13,14,20]. Remarkably, subclinical CMV exposure and reactivation lead not only to long-term CD57+CD4+ memory T cells [35] but also to a higher proportion of CD4+CD28null T cells, which correlates with a reduced response to vaccination [36]. These data may confirm our findings showing a positive correlation between serum IgG levels for CMV and the proportion of senescent T cells, defined as CD28−CD57+ (for both CD4+ and CD8+ T cells), in the Allele-1 group following vaccination, and also allow us to speculate that these associations may influence and lead to an altered COVID-19 immune response encompassing both SARS-CoV-2 infection and vaccination.

Other important findings were the positive associations between serum levels of IgG for CMV and some pro-inflammatory cytokines such as IL-17 and IFN-γ in the pre-vaccination period and TNF-α in the post-vaccination period. In this sense, it has been reported that CMV DNAemia was positively correlated with elevated serum IL-17 levels and that Th17 cells activated in this context also co-expressed IFN-γ and/or TNF-α [37]. Furthermore, the positive correlation between serum IgG levels for CMV and the ratios of IL-17/IL-10, IFN-γ/IL-10, and IL-6/IL-10 supports the literature claiming that the chronic presence of CMV can induce a pro-inflammatory state [14,38], which, as previously mentioned, may impair the immune response to vaccination in older adults [13,14,20].

Similarly, the combination of the higher percentage of intermediate monocytes, which can release higher pro-inflammatory cytokine levels in response to microbial products [39], with the reduced percentage of non-classical monocytes, which have anti-inflammatory properties, could corroborate this suggestion that the Allele-1 group has a systemic pro-inflammatory status that may presumably affect the COVID-19 vaccination response. According to the literature, there are not only the number and activation of all subtypes of monocytes unaffected by CMV serostatus [40,41], which is consistent with our data, but also evidence of an increase in the number of all subtypes of monocytes with increasing age and regardless of gender [40], particularly the intermediate subtype, which alters the pattern of circulating monocytes [42], as observed here. In addition, CMV itself can stimulate monocytes in a latent form, which can lead to an increase in circulating inflammatory cytokine levels [41,43]. In particular, intermediate monocytes play a central role in antigen presentation due to their higher expression of MHC class II and some costimulatory molecules, especially CD86 [44]. Based on this information, we can hypothesize that the increased rates of intermediate monocytes found in the Allele-1 group after vaccination are likely related to the presentation of CMV antigens, which could contribute to generate a systemic pro-inflammatory state that favors this virus and consequently leads to an altered immune response to SARS-CoV-2 infection and/or vaccination, as previously reported [20].

Although the Alleles-1/2 group also had a lower proportion of non-classical monocytes after vaccination, these heterozygous individuals showed a significant negative correlation between the IgG-specific serum levels for SARS-CoV-2 antigens and the intermediate monocyte rates before vaccination, similar to the Allele-1 group. This last finding may demonstrate that an appropriate immune response to SARS-CoV-2 infection occurred in a regulated inflammatory state and may confirm our suggestion that the increased percentage of this pro-inflammatory monocyte subtype observed in the Allele-1 group may favor a systemic pro-inflammatory status that could impair the immune response to SARS-CoV-2 infection and vaccination in older adults who are CMV seropositive.

Corroborating this suggestion that balanced inflammation may favor the immune response to COVID-19 vaccination, a positive correlation was demonstrated between circulating IgG levels specific for SARS-CoV-2 antigens and IL-10 in both the Allele-2 and Allele-1/2 groups. In particular, IL-10 has significant anti-inflammatory properties, and our group has previously documented that older adults who showed a better antibody response to influenza virus vaccination not only showed a positive correlation with circulating IL-10 levels but also a negative correlation between this cytokine and IgG specific for CMV [28]. Regarding the latter observation, although the Alleles-1/2 group did not show a correlation between serum levels of IgG specific for SARS-CoV-2 and CMV, the Allele-2 group did. These results suggest that the presence of a single C allele in the gene IFNλ 3/4 may partially influence the immune response to SARS-CoV-2 and CMV, as the Allele-2 group, homozygous for T/T alleles, showed that the better antibody response to infection and vaccination against SARS-CoV-2 was negatively associated with CMV-seropositivity.

A lower incidence of active CMV infections and a trend towards lower replication of this virus in individuals with the TT genotype than in those with the CC genotype have been described in the literature, suggesting a protective effect of the T allele (rs12979860) against CMV infection and reactivation [45]. In this sense, it has been hypothesized that this protective effect of the T allele could be due at least to robust CMV-specific T cell responses [46]. Indeed, our previous data confirm this assumption, as the better response to influenza virus vaccination in CMV-seropositive older adults was associated with both a systemic anti-inflammatory status and an enhancement of naïve CD8+ T cells [28]. Although we did not observe any associations between the antibody response to COVID-19 and naïve T cells, there was a significant negative correlation between serum IgG-specific levels for SARS-CoV-2 antigens and senescent CD8+ T cells in the Allele-2 group, which may support the notion that the TT genotype exerts a protective effect through robust CMV-specific T cell responses, as the presence and maintenance of senescent T cells, particularly CD8+ T cells, is a hallmark of chronic CMV infection and also impairs the immune response to vaccination in older adults [20,47].

Consistent with this suggestion that the TT genotype provides better protection against viral infections, preferably CMV, the Allele-2 group had higher circulating IFN-β levels and an increased IFN-β/IL-10 ratio than other groups during the study periods. In addition, significant positive correlations between IgG levels specific for SARS-CoV-2 and IFN-α2 or IFN-α2/IL-10 ratio at pre-vaccination were demonstrated in this group. It is generally recognized that type I IFNs, such as IFN-α and IFN-β, play a central role in the immune response to viral infections, including SARS-CoV-2 and CMV [48]. In this context, the dysregulated innate immune response described in severe COVID-19 has been associated with both a limited and a delayed type I IFN response [49], and it has also been shown that type I IFNs can reduce CMV replication in macrophage-like cells [50,51]. Interestingly, a combination of IFNλ and type I IFN showed synergistic anti-HCV activity in vitro [52], and it has been suggested that IFNλ can fine-tune type I IFN levels and act at optimal concentrations that control viral replication [51,53]. This information may suggest that not only the higher IFN-β levels observed in the Allele-2 group may be useful for controlling CMV infection but also that the positive correlation between IgG levels specific for SARS-CoV-2 and IFN-α2 or the IFN-α2/IL-10 ratio may benefit the immune response to viral infection and vaccination, demonstrating a putative link between the IFNλ polymorphism and type I IFNs in the study. Furthermore, the negative association between serum IgG-specific levels for SARS-CoV-2 and BMI or age, which are considered pro-inflammatory triggers, found in the Allele-1 and Alleles-1/2 groups, respectively, support the suggestion that a systemic pro-inflammatory status may promote an immune system imbalance that impairs the response to SARS-CoV-2 infection [54,55] as well as the response to vaccination against pathogens, including SARS-CoV-2 [55].

Lastly, some limitations of the present study are highlighted, such as: (i) the discrepant number of volunteers between groups according to genotyping, 20 participants showed homozygosity for the C/C alleles (Allele-1 group), 5 showed homozygosity for the T/T alleles (Allele-2 group) and 17 showed heterozygosity (C/T group, Alleles-1/2). It is worth noting that this slight discrepancy does not invalidate the results presented or affect the statistical significance of this study. Considering that the frequency of alleles reflects the genetic variability of the expected population and, in addition, confirms our previous study in which the number of older adults who had genotypes with cytosine (C/C and C/T) was greater than with thymine only (T/T) [20]; (ii) the impossibility of performing a statistical analysis when the volunteers were separated into subgroups of “responders” and “non-responders”, in agreement with the data obtained in the VNT; (iii) the impossibility of conducting future analyses regarding the impact of the polymorphism for IFNλ on immunogenicity in homologous vaccine models, both for other vaccines against COVID-19 and for the third dose of CoronaVac, given the introduction of a heterologous vaccination schedule in the country; (iv) the lack of data not only regarding the analysis of other polymorphisms in IFNλ genes, such as IFNλ4 rs11322783, which may also play a role in COVID-19 context; (v) but also regarding serum levels of IFNλ, which unfortunately could not be determined during the development of the study; and (vi) the lack of comparison with CMV-seronegative older adults immunized with the first two doses of the CoronaVac vaccine. Although we understand that this comparison could strengthen the meaning of our study, it is important to emphasize that it is well known that CMV infection rates increase with age [56], reaching an overall seroprevalence of approximately 90% [57], making it difficult to include a group of older CMV-seronegative individuals.

## 5. Conclusions

Overall, the observation that the Allele-1 (C/C) group had higher total serum levels of specific IgG for SARS-CoV-2 antigens post-vaccination not only corroborates our previous findings that individuals with Allele-1 for the gene IFNλ may have differences in immune response to the virus and/or vaccination compared to individuals with Al-lele-2 or Alleles-1/2, but also the literature suggesting that pro-inflammatory system status may be due to the lower control of CMV infection that occurs in individuals with this allelic discrimination.

Although our results can putatively suggest that the assessment of IFNλ genotype and CMV seropositivity may be useful to identify individuals at higher risk for inadequate immune response to COVID-19 vaccination in the older adult population, further studies need to be conducted to improve our understanding of this scenario.

## Figures and Tables

**Figure 1 vaccines-13-00785-f001:**
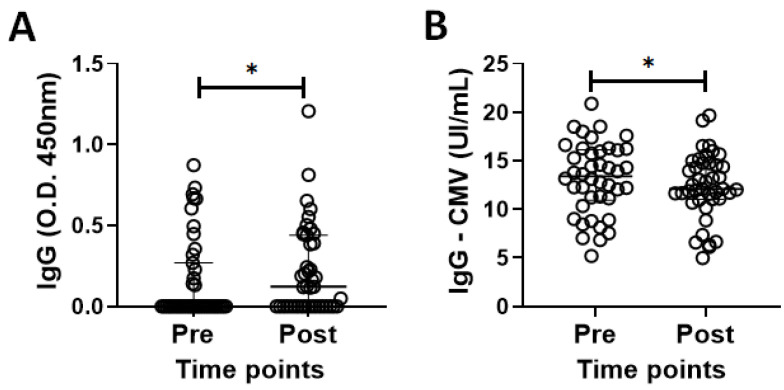
Total serum levels of specific IgG for SARS-CoV-2 (O.D. 450 nm—(**A**)) and CMV (UI/mL—(**B**)), in the volunteer group together, both before (pre) and 30 days after administration of the second dose of CoronaVac vaccine (post). Values are expressed as median and interquartile range. * indicates *p* < 0.05.

**Figure 2 vaccines-13-00785-f002:**
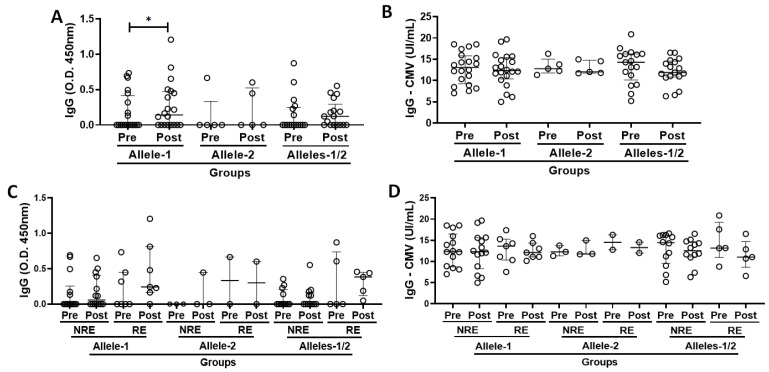
Total serum levels of specific IgG for SARS-CoV-2 (O.D. 450 nm—(**A**)) and CMV (UI/mL—(**B**)) when the volunteers were categorized according to IFNλ 3/4 gene polymorphism [Allele-1 (C/C), Allele-2 (T/T), and Alleles-1 2 (heterozygous, C/T), both before (pre) and 30 days after administration of the second dose of CoronaVac vaccine (post). In addition, in (**C**,**D**), the total serum levels of specific IgG for SARS-CoV-2 (O.D. 450 nm) and CMV (UI/mL) in the volunteers grouped not only according to IFNλ 3/4 gene polymorphism [Allele-1 (C/C), Allele-2 (T/T), and Alleles-1/2 (heterozygous, C/T)] but also according to the presence (RE, responder) or absence (NRE, non-responder) of neutralizing antibodies against SARS-CoV-2. Values are expressed as median and interquartile range. * indicates *p* < 0.05.

**Figure 3 vaccines-13-00785-f003:**
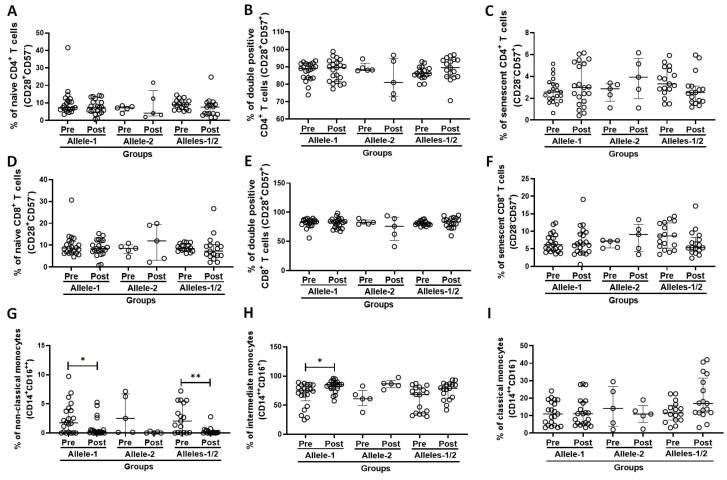
Percentage of CD4 = T cells (naive—CD28+CD57−, (**A**); double-positive—CD28+CD57+, (**B**); and senescent—CD28-CD57+, (**C**); CD8+ cells (naïve—CD28+CD57−, (**D**); double-positive—CD28+CD57+, (**E**); and senescent—CD28-CD57+, (**F**); as well as the percentage of monocyte subtypes (non-classical—CD14+CD16++, (**G**); intermediate—CD14++CD16+, (**H**); and classical—CD14++CD16−, (**I**); both before (pre) and 30 days after the administration of the second dose of CoronaVac vaccine (post), in the volunteers categorized according to polymorphism in the IFNλ 3/4 gene [Allele-1 (homozygous, C/C), Allele-2 (homozygous, T/T), and Allele-1;2 (heterozygous, C/T)]. Values are expressed as median and interquartile range. * indicates *p* < 0.05 and ** indicates *p* < 0.01.

**Figure 4 vaccines-13-00785-f004:**
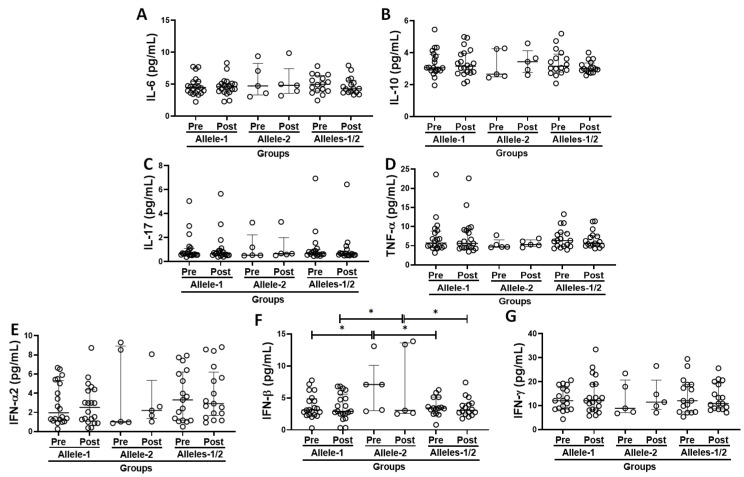
Circulating cytokine levels of IL-6 (**A**), IL-10 (**B**), IL-17 (**C**), TNF-α (**D**), IFN-α2 (**E**), IFN-β (**F**), and IFN-γ (**G**), expressed as median and interquartile range, both before (pre) and 30 days after administration of the second dose of CoronaVac vaccine (post), in the volunteers categorized according to polymorphism in IFNλ 3/4 [Allele-1 (homozygous, C/C), Allele-2 (homozygous, T/T), and Allele-1;2 (heterozygous, C/T)]. * indicates *p* < 0.05.

**Figure 5 vaccines-13-00785-f005:**
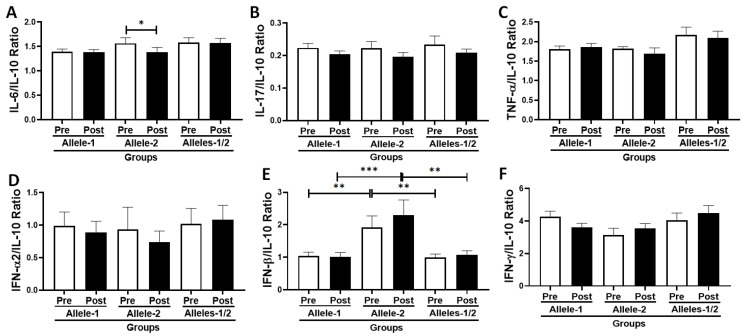
Ratio between the levels of pro-/anti-inflammatory cytokines [IL-6/IL-10 (**A**), IL-17/IL-10 (**B**), TNF-α/IL-10 (**C**), IFN-α2/IL-10 (**D**), IFN-β/IL-10 (**E**), and IFN-γ/IL-10 (**F**), expressed as mean and standard deviation, both before (pre) and 30 days after administration of the second dose of CoronaVac vaccine (post) in the volunteers categorized according to polymorphism in IFNλ 3/4 [Allele-1 (homozygous, C/C), Allele-2 (homozygous, T/T), and Allele-1;2 (heterozygous, C/T)]. * indicates *p* < 0.05. ** indicates *p* < 0.01. *** indicates *p* < 0.001.

**Table 1 vaccines-13-00785-t001:** Demographic and anthropometric data, presented as mean and standard deviation (X ± SD), of the participants in the present study, both overall and grouped by allelic discrimination, who were immunized with two doses of the CoronaVac vaccine against COVID-19.

Parameters	Volunteers				*p* Value
	Total (*n* = 42)	Allele-1 (C/C) (*n* = 20)	Allele-2 (T/T) (*n* = 05)	Alleles-1/2 (C/T) (*n* = 17)	
Age (years)	73.7 ± 4.5	73.9 ± 4.9	72.2 ± 4.4	74.0 ± 4.0	0.722
Older men (n)	12	4	2	6	0.439
Older women (n)	30	16	3	11	
Ratio M–W	1:2.58	1:4.25	1:1.5	1:1.83	
Height (m)	1.58 ± 0.10	1.56 ± 0.10	1.61 ± 0.10	1.59 ± 0.12	0.488
Weight (kg)	64.4 ± 12.7	63.6 ± 12.0	62.9 ± 14.3	65.8 ± 13.6	0.839
BMI * (kg/m^2^)	25.7 ± 4.4	26.1 ± 5.0	23.8 ± 3.7	25.7 ± 3.8	0.579

* BMI—Body mass index.

**Table 2 vaccines-13-00785-t002:** Analysis of Spearman correlation coefficients between all parameters evaluated in the present study before (pre-vaccination) and 30 days after administration of the second dose of CoronaVac vaccine (post-vaccination) in subjects grouped according to allelic differentiation into Allele-1 (C/C), Allele-2 (T/T), and Alleles-1/2 (heterozygous, C/T).

Allele-1 group (C/C)					
Parameters	Pre-vaccination		Parameters	Post-vaccination	
	*rho*-value	*p*-value		*rho*-value	*p*-value
IgG-COVID-19 X BMI *	−0.450	0.040	IgG-CMV X IFN-γ	0.554	0.009
IgG-CMV X IL-17	0.558	0.008	IgG-CMV X IL-6/IL-10	0.475	0.029
IgG-CMV X IFN-γ	0.534	0.012	IgG-CMV X IFN-γ/IL-10	0.697	<0.001
IgG-CMV X IL-17/IL-10	0.510	0.018	IgG-CMV X senescent CD4+ T cells	0.655	<0.001
IgG-CMV X IFN-γ/IL-10	0.606	0.003	IgG-CMV X senescent CD8+ T cells	0.675	<0.001
Allele-2 group (T/T)					
Parameters	Pre-vaccination		Parameters	Post-vaccination	
	*rho*-value	*p*-value		*rho*-value	*p*-value
IgG-COVID-19 X TNF-α	−0.995	<0.001	IgG-COVID-19 X IL-10	0.878	0.048
IgG-COVID-19 X IFN-α2	0.996	<0.001	IgG-COVID-19 X senescent CD8+ T cells	−0.886	0.045
IgG-COVID-19 X IFN-γ	0.918	0.027			
IgG-COVID-19 X IL-17	−0.968	0.006			
IgG-COVID-19 X TNF-α/IL-10	−0.999	<0.001			
IgG-COVID-19 X IFN-α2/IL-10	0.998	<0.001			
IgG-COVID-19 X IFN-γ/IL-10	0.922	0.025			
IgG-COVID-19 X IL-17/IL-10	−0.990	0.001			
Alleles-1/2 group (C/T)					
Parameters	Pre-vaccination		Parameters	Post-vaccination	
	*rho*-value	*p*-value		*rho*-value	*p*-value
IgG-COVID-19 X Intermediate	−0.528	0.029	IgG-COVID-19 X IL-10	0.513	0.035
			IgG-COVID-19 X IFN-γ	0.567	0.017

* BMI = body mass index.

**Table 3 vaccines-13-00785-t003:** Results of multivariate regression analysis adjusted for serum IgG-specific levels for SARS-CoV-2 antigens before (pre) and 30 days after (post) administration of the second dose of CoronaVac vaccine for COVID-19 in the volunteer groups.

IgG-Specific for SARS-CoV-2 Antigen Adjusted								
Parameters	Allele-1 group (C/C)							
	Pre-vaccination				Post-vaccination			
	*β*-value	95% CI *	*p*-value	*R* ^2^	*β*-value	95% CI *	*p*-value	*R* ^2^
BMI **	−0.03636	−0.014 to 0.042	0.0433	0.5496	−0.03849	−0.067 to −0.0097	0.0124	0.1686
IL-17	Ns	Ns	Ns	Ns	−0.1911	−0.359 to −0.023	0.0301	0.6288
IL-17/IL-10	Ns	Ns	Ns	Ns	−0.6557	−1.272 to −0.039	0.0392	0.5796
IFN-γ	Ns	Ns	Ns	Ns	0.00445	0.002 to 0.007	0.0049	0.8339
IFN-γ/IL-10	Ns	Ns	Ns	Ns	0.01261	0.004 to 0.022	0.0102	0.6984
Parameters	Allele-2 group (T/T)							
	Pre-vaccination				Post-vaccination			
	*β*-value	95% CI *	*p*-value	*R* ^2^	*β*-value	95% CI *	*p*-value	*R* ^2^
IgG-CMV	−0.0354	−0.033 to −0.038	0.0001	0.9998	−0.2331	−0.273 to −0.193	0.0003	0.9997
Non classic	Ns	Ns	Ns	Ns	6.734	3.774 to 9.69	0.0054	0.9996
Parameters	Alleles-1/2 group (C/T)							
	Pre-vaccination				Post-vaccination			
	*β*-value	95% CI *	*p*-value	*R* ^2^	*β*-value	95% CI *	*p*-value	*R* ^2^
Age	Ns	Ns	Ns	Ns	−0.02826	−0.059 to −0.002	0.0379	0.2423

* CI = coefficient interval; ** BMI = body mass index; Ns = not significant.

## Data Availability

All data included in the present study are available from the corresponding authors upon reasonable request.

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
