# Peer review of "Polymorphism in IFNλ Can Impact the Immune/Inflammatory Response to COVID-19 Vaccination in Older CMV-Seropositive Adults"

_vaccines, 2025, doi:10.3390/vaccines13080785_

Round 1

Reviewer 1 Report

Comments and Suggestions for Authors

The article presented for review focuses on studying the influence of IFN-lambda gene polymorphism on the development of a response to the COVD-19 vaccination in CMV-seropositive elderly people. The article has several shortcomings that make it difficult to understand.

1) Studying the surface phenotype of cells after freezing and defrosting does not seem entirely correct. During the thawing process, cells lose a number of surface markers. For each cell subset, an illustration of gating is essential.

2) It would be better to add explanations to the figure’s legends. Most of the figures have pre- and post-abbreviations, but the legends don't explain what they mean. The figures do not show any significant changes. This makes it difficult to understand the article.

3) The results (3.3) do not explain what gene alleles are. What are the Allele 1, 2, and 3? Explanations should be added.

4) The text of the article refers to the availability of supplementary materials (Supplementary Materials - Fig S1), but these data are not available.

5) From Materials and Methods it is not clear whether patients have provided their informed consent.

Author Response

We would like to inform you that all responses to the reviewer's comments/suggestions/questions are provided in the attached PDF file.

Reviewer 2 Report

Comments and Suggestions for Authors

The aim of the study is to investigate how cytomegalovirus (CMV) seropositivity and genetic variation in the IFNλ3/4 gene (rs12979860 polymorphism) influence the immune response specifically SARS-CoV-2-specific IgG antibody production and inflammatory markers following administration of two doses of the CoronaVac COVID-19 vaccine in aged individuals.

Please ensure that all figure panels are presented in proper alphabetical order. For instance, in Figure 3, the sequence begins with panel 3A, followed by 3D, then 3B and 3E, and subsequently 3C and 3F. This disordered presentation may confuse readers. It is recommended that figure panels be arranged and labeled sequentially (e.g., 3A, 3B, 3C, etc.) to enhance clarity and readability.

Figure 1 presents IgG OD450 data and the dynamics of IgG levels before and after vaccination in the CMV-seropositive group. However, the distinction between Figures 1B and 1C is unclear—do they display different datasets, or is the same dataset presented using a different format or statistical treatment? A similar concern applies to Figure 2C. Please clarify this in the figure legends and explicitly describe the statistical tests applied in each panel to facilitate interpretation.

While the study presents intriguing correlations between IFNλ3/4 gene polymorphisms, systemic inflammatory markers, and vaccine-induced IgG responses in CMV-seropositive older adults, the interpretation of immunogenicity outcomes must be approached with caution. Notably, neutralizing antibody data, which serve as a more direct correlate of protective immunity, were either not presented or not clearly interpreted. Furthermore, the study lacks direct evidence of CMV reactivation, such as CMV DNAemia or viral load measurements. Please address limition of your study, and number of participants especially in Allele -2 (n=5) is low for power. These should be considered, and conclusion should be revised addressing these points. 

Given these limitations—particularly the absence of functional neutralization assays and virological confirmation of CMV activity—the conclusions regarding altered immune responses and the immunomodulatory impact of CMV serostatus remain speculative. Future studies should incorporate rigorous immunological and virological assessments, including neutralization assays and CMV reactivation monitoring, to substantiate the proposed mechanisms.

Additionally, the overall quality and resolution of the figures should be improved to meet publication standards and support accurate interpretation.

No supplementary figures have been found in the submitted manuscript. Please ensure that all data referenced in the draft, including supplementary materials, are properly attached. Additionally, please include a detailed description of the gating strategies in the supplementary methods section.

Please review the manuscript carefully for correct usage of punctuation, particularly periods (.) and parentheses (). In several instances, periods are either missing at the end of sentences or are used redundantly. Some parentheses are left unclosed after being opened. 

Comments on the Quality of English Language

It requires a grammar check before publication. 

Author Response

(The authors gave the same response as above.)

Round 2

Reviewer 1 Report

Comments and Suggestions for Authors

All comments were considered by the authors

Reviewer 2 Report

Comments and Suggestions for Authors

Thanks to authors for addressing my concerns.